# Cortical Spreading Depolarization and Delayed Cerebral Ischemia; Rethinking Secondary Neurological Injury in Subarachnoid Hemorrhage

**DOI:** 10.3390/ijms24129883

**Published:** 2023-06-08

**Authors:** Ashir Mehra, Francisco Gomez, Holly Bischof, Daniel Diedrich, Krzysztof Laudanski

**Affiliations:** 1Department of Neurology, University of Missouri, Columbia, MO 65212, USA; fegyr7@umsystem.edu; 2Penn Presbyterian Medical Center, Philadelphia, PA 19104, USA; holly.bischof@pennmedicine.upenn.edu; 3Department of Anesthesiology and Perioperative Care, Mayo Clinic, Rochester, MN 55905, USA; daniel.diedrich@mayo.edu (D.D.);

**Keywords:** SAH, CSD, neuroinflammation, delayed cerebral ischemia

## Abstract

Poor outcomes in Subarachnoid Hemorrhage (SAH) are in part due to a unique form of secondary neurological injury known as Delayed Cerebral Ischemia (DCI). DCI is characterized by new neurological insults that continue to occur beyond 72 h after the onset of the hemorrhage. Historically, it was thought to be a consequence of hypoperfusion in the setting of vasospasm. However, DCI was found to occur even in the absence of radiographic evidence of vasospasm. More recent evidence indicates that catastrophic ionic disruptions known as Cortical Spreading Depolarizations (CSD) may be the culprits of DCI. CSDs occur in otherwise healthy brain tissue even without demonstrable vasospasm. Furthermore, CSDs often trigger a complex interplay of neuroinflammation, microthrombi formation, and vasoconstriction. CSDs may therefore represent measurable and modifiable prognostic factors in the prevention and treatment of DCI. Although Ketamine and Nimodipine have shown promise in the treatment and prevention of CSDs in SAH, further research is needed to determine the therapeutic potential of these as well as other agents.

## 1. Introduction

Subarachnoid Hemorrhage (SAH) refers to the extravasation of blood between the pia and the arachnoid mater and is typically caused by an aneurysm rupture or traumatic brain injury. It is a life-threatening condition with an annual incidence of roughly 9 per 100,000 in the United States and 7.9 per 100,000 worldwide [1]. With only 60% of patients returning to their premorbid lifestyles [2,3], SAH is notorious for poor outcomes and is associated with a high likelihood of disability, cognitive impairment, and death [1,3]. 

Poor outcomes in SAH are partly due to its unique ability to induce novel neurological injury days after the onset of the hemorrhage. As such, the neurological injury in SAH has been described to occur in two phases. ‘Early Brain Injury’ refers to neurological damage that occurs within 72 h of the onset of SAH, while ‘Delayed Cerebral Ischemia’ (DCI) is a form of secondary injury that refers to any additional insult occurring beyond 72 h of the index hemorrhage [4]. Clinically, DCI manifests as new onset altered mentation or focal neurological deficits lasting more than one hour or the appearance of a new ischemic lesion on computed tomography (CT) or magnetic resonance imaging (MRI) [5]. DCI accounts for up to one-third of new deficits and deaths subsequent to SAH, with an incidence of up to 1.2 million patients per year worldwide [6,7].

DCI may be avoidable and reversible [4]. Historically, it was noted that up to 70% of SAH patients showed evidence of delayed radiographic vasoconstriction. Delayed onset vasospasm was therefore thought to be the culprit of DCI [8]. Unfortunately, treatments focusing solely on the reversal of observable vasospasm were incompletely effective. For example, Clazosentan, an endothelin receptor antagonist that reduced vasospasm in a dose-dependent manner, failed to alleviate DCI. More recently, DCI has been recognized to develop in the absence of vasospasm [5,9]. Transcranial Doppler (TCD) and radiographic studies showed vasospasm may only have a positive predictive value in DCI of up to 57% [10]. These findings indicate that DCI is a multifactorial phenomenon [11,12,13,14].

Cortical Spreading Depolarizations (CSDs) are a pathological phenomenon characterized by profound depolarizations with loss of transmembrane potential energy. CSDs occur in SAH, often in even absence of demonstrated vasospasm, and are closely interlinked to neuroinflammation [5]. A complex interplay between cortical spreading depolarization, neuroinflammation, and microthrombi formation has been increasingly recognized to play a key role in DCI (Figure 1) [15].

## 2. Materials and Methods

A search of the PubMed database [1971 to 20 January 2023] was conducted using the search terms “cortical spreading depolarization”, “spreading depression”, “subarachnoid hemorrhage”, and “hemorrhagic stroke”. Duplicates and publications not published in English were excluded. One hundred eighty-nine abstracts were initially identified, and FG, AM, and HB reviewed them for relevancy. FG was the ultimate arbiter for inclusion. Secondary papers and prior reviews were included as relevant to the readers.

## 3. Pathophysiology of Cortical Spreading Depolarization

CSDs are self-propagating waves of pathological depolarization afflicting neurons and glial cells [16,17], manifesting in various CNS pathologies such as stroke, traumatic brain injury, and SAH [2]. As CSDs transverse through otherwise healthy brain tissue, they lead to metabolic deficits [18] while mediating an interplay of the neurological, vascular, and immune systems [19,20,21,22,23]. These mechanisms are intertwined and self-propagating. CSDs have been described as the paramount secondary neuronal injury mechanism and are considered a marker of neuronal dysfunction in gray matter [24,25].

CSDs characteristically disrupt membrane ion trafficking by causing cation influx, i.e., Na+, Ca++, at a rate that overwhelms Na-K ATPase-dependent efflux (Figure 2). Cytotoxic edema ensues, and roughly 90% of the potential energy contained in neuronal transmembrane gradients is lost [26]. Excitotoxicity follows via a positive feedback loop wherein N-methly-D-aspartate (NMDA) receptor activation leads to self-sustaining net inward cation (Ca^2+^ and Na+) currents that cause cellular osmotic and metabolic derangements, disruption of second messenger signaling, and Ca^2+^ dysregulation [27,28].

As cells are injured, extracellular concentrations of glutamate and K+ rise sharply via mechanisms including neuronal release and impaired astrocytic glutamate clearance from the extracellular fluid (ECF). Reversed glutamate transport sees astrocytes release glutamate instead of clearing glutamate from the ECF [3,29,30,31,32,33,34]. Glutamate released from injured or lysed cells eventually washes over nearby survivors to start this injurious cycle anew [8,27,35]. As ECF glutamate and potassium concentrations continue to rise, simultaneous opening of multiple channel types and marked conductance rises occur with CSDs. ATP-dependent, voltage-gated, iontophoretic, and hemichannels are activated [8,36], leading to further excitotoxicity, neuroinflammation, and apoptosis [37].

### Spreading Depression and Terminal Depolarizations

CSDs typically spread omnidirectionally from the lesioned area at an average speed of 2–5 mm/min and last for a minimum of 15 s. CSDs are followed by a period of depressed electrocorticographic (ECoG) activity, a phenomenon termed Spreading Depression (Figure 3) [19,21,22,23]. This period of electrical inactivity is mediated by a loss of neuronal electric potential due to profound depolarization, leading to a transient inability to produce action potentials, not dissimilar to a refractory period [18]. While some cells will recover spontaneously from CSDs, “terminal depolarizations” are often seen wherein the neurons fail to recover and subsequently die. These neurons are not able to repolarize due to insurmountable damage, as seen in severe persistent hypoxia, hypoglycemia, or exposure to high concentrations of K+, glutamate, or the Na+ pump inhibitor ouabain. The main mediators of this apoptosis include increased intracellular Ca++, activation of intracellular proteases, and excessive free radical production [2,38].

CSDs are thus implicated as causing DCI from protracted or failed cellular recovery and contributing to patient morbidity and mortality [14]. Profound metabolic deficits are generated and exacerbated by CSDs [39], converting the CSD into a wave of terminal depolarization.

## 4. Cortical Spreading Depolarization and the Failure of Vascular Autoregulation

CSDs cause microvascular failure. Under physiological conditions, local blood flow is tailored to local metabolic demand by a network of neurons, interneurons, astrocytes, basal laminae, and smooth muscle cells, termed neurovascular units (NVU). The curated balance of metabolic supply and demand is termed neurovascular coupling [2,40,41]. As metabolic demand increases, so does the vascular blood flow.

CSDs are associated with elevated extracellular potassium, which directly depolarizes vascular smooth muscle and thereby promotes vasoconstriction [42]. Additionally, CSDs promote inhibition of the local production of nitric oxide, a potent vasodilator. A combination of these two factors results in a paradoxical reduction in local blood flow despite an increase in metabolic demand, a phenomenon referred to as “inverse neurovascular coupling” or simply “neurovascular uncoupling” [2,41]. The recovery from spreading depolarization and its subsequent depression is delayed due to the aforementioned inverse neurovascular coupling [2]. In animal models, prolonged SD, along with inverse neurovascular coupling, proved sufficient to cause widespread necrosis [43].

Finally, CSDs exacerbate reactive oxygen species (ROS) production in activated microglia [44,45]. As ROS accumulate, peroxidation of membrane lipids and proteins impairs smooth muscle ion channels, thus reducing myogenic reactivity [46,47]. Concurrently, astrocytic endfeet edema may lead to microvascular compression, further impairing vascular reactivity [48].

## 5. Cortical Spreading Depolarization in Delayed Cerebral Ischemia

CSDs incidence in SAH patients is 72–80% [3,21] and has been shown to be independent of radiographic evidence of vasospasm [24,49,50]. CSDs in SAH have demonstrated biphasic peaking at days 0 and 7 after the onset of the hemorrhage [21]. They have also shown a temporal relation with DCI, as one study demonstrated that 75% of all recorded CSD occurred between the fifth and seventh day after the onset of SAH [51]. Animal models have shown prolonged CSDs in concert with inverse neurovascular coupling are sufficient to cause widespread necrosis [43]. Murine aneurysmal SAH models have further demonstrated that CSD led to ischemic lesions as identified by MRI both ipsilateral and contralateral to the ruptured vessel [52].

### 5.1. Factors Affecting CSD in SAH

Factors affecting the development of CSDs in SAH include the volume of the hemorrhage, location of the hemorrhage, time since the onset of the hemorrhage, and the presence of hemoglobin degradation products.

As expected, the volume of subarachnoid hemorrhage directly correlates with the occurrence of CSDs [3,53,54,55]. That being said, swine models demonstrated that even a focal accumulation of subarachnoid blood was sufficient to trigger the development of both CSD clusters and early infarcts within 72 h of ictus. The study recorded CSDs in 76% of experimental animals and reported clusters of depolarization in 41%. Although research regarding affiliations between the location of the SAH and CSD is limited, sulcal clots have been shown to have a high risk of CSDs [24,25].

The time since the onset of SAH also appears to be a factor in the development of CSD in SAH. A time-locked correlation between the development of CSDs and hemoglobin degradation products has been found, thereby possibly explaining the delayed nature of CSD-mediated neurological insult in SAH [56]. A feline study demonstrated that hemolyzed red blood cells injected into the subarachnoid space were more likely to cause suppression of activity on electroencephalogram (EEG) than whole blood, packed autologous red blood cells, serum, or crystalloid preparations [57]. CSDs are implicated as the true cause of the delayed second injury secondary to the ischemia they create [14].

### 5.2. Relation between CSD and the Size of DCI

DCI lesion growth has been demonstrated in animal models to be greater in the presence of CSDs, noting (241 ± 233 mm^3^) as opposed to controls (29 ± 54 mm^3^) (*p* = 0.001) [58]. However, further murine studies showed CSDs would only induce irreversible injury in at-risk tissue [59]. In one clinical trial enrolling 50 patients, a closer correlation was found between CSDs and DCI than that of angiographic vasospasm or perfusion CT scanning (odds ratio 2.064, 95% confidence interval 1.045–4.075, *p* = 0.037, the area under the curve 0.836) [60].

### 5.3. Measuring CSDs

A study employing cortical electrodes reported 298 CSD waves in 13 of 18 (72%) patients, wherein 7 patients developed DCI. CSD, therefore, had a positive predictive value of 86% and a negative predictive value of 100% [13,61,62]. Another study showed peak total depression time per day; that is, the total electrographic silence as measured via cortical electrodes was higher in SAH patients with poor outcomes while showing a correlation between CSDs and transient drops in tissular oxygen [63].

The currently used method of TCD may be inferior as metanalysis demonstrated a positive predictive value of 57% (95% CI 38–71%) and a negative predictive value of 92% (95% CI 83–96%) [10]. The noninvasive, cost-effective, and easily accessible nature of TCD allows it to be the standard of care for monitoring currently. Moving forward, cortical electrode placement may be a more accurate way of assessing DCI in patients undergoing operative management of their SAH at larger tertiary centers.

## 6. The Interplay between Cortical Spreading Depolarization and Neuroinflammation

A randomized controlled trial evaluated the interplay between CSDs, vasospasm, and microcirculation in the setting of DCI. The authors used digital subtraction angiography to measure proximal and peripheral cerebral circulation time 7–11 days after the onset of SAH. Multivariate analysis showed the number of CSDs on the day of angiography was the only significant factor leading to DCI [24,60]. It is important to recognize that microglial activation and neuroinflammation play a key role in the appearance and spread of CSDs. A few cellular and molecular mediators of this interplay are discussed herein.

### 6.1. Microglia

Microglia are a heterogenous group of resident macrophages that act as the primary effector of the immune system in the CNS. This cell line orchestrates neuroinflammation by secretion of interleukins 1, 6, 18, and 23, as well as TNF-alpha in response to diverse stimuli, including SAH [64]. While several variants of activated microglia phenotypes have been described, tumor necrosis factor-α and interleukin-6 (IL-6) release is associated with the M1 subtype. Other evidence of microglial activation in SAH has been described, including increased CSF of microglial oxygenase 1 activity and peroxiredoxin 2 and increased major histocompatibility complex 2 expression [65].

Microglia may be activated by damage-associated molecular patterns (DAMPs), including proinflammatory cytokines, cellular debris, and hemoglobin degradation products that are present in subarachnoid blood [66]. Additionally, CSDs are posited to directly activate microglia. Murine CSD models have demonstrated upregulation of proinflammatory genes (Cd53, Ms4a6d, Anxa2, Ccl2, Vim, C3ar1, and Timp1) with a skew towards IFN binding sites [67]. The latter is implicated in the regulation of blood-brain barrier permeability [67]. Another murine model demonstrated increased IL-1, CCL2, and TNF-alpha with less notable increases in IL-6 and ICAM-1 ipsilateral to induced CSDs. These pro-inflammatory cytokines were shown to persist for up to 50 h after the resolution of CSDs [68]. ICAM contributes to neuroinflammation via the promotion of neutrophil and monocyte adhesion and migration [22].

Microglial activation has been noted to contribute to cell death after SAH [69]. Notably, one elegant study on microglia-depleted hippocampal slices demonstrated no CSD responses vs. control [70]. Thus, although neuroinflammation is an increasingly recognized phenomenon resulting from SAH [71], it may also play a role in CSD genesis. Recent small animal experiments on microglial suppression via PLX3397 decreased microglia cell accumulation and COX2 gene expression; however, given said substances’ effects on peripheral immunity, implicated mechanisms remain to be fully elucidated [69].

### 6.2. Astrocytes

Astrocytes are part of the cerebral glia, tasked with the regulation of synaptic transmission, NVU function, and maintenance of blood-brain barrier integrity via their endfeet (Figure 4) [72,73,74]. Under physiological conditions, they inhibit CSD initiation via K+ buffering through Na-K ATPase and glutamate uptake through GLT-1 receptors [53]. However, the astrocytic response may be deleterious in a pro-inflammatory setting.

SAH patients exhibit high levels of circulating and CSF glial fibrillary protein, a marker of astrocyte activation [74]. Activated astrocytes potentiate neuroinflammation via the secretion of IL-1, IL-6, TNF-alpha, and Interferon-gamma. Additionally, they produce metalloproteinase 9 (MMP-9), which degrades numerous components of the blood-brain barrier and thereby allows for cerebral edema [73,74].

CSD-exposed astrocytes exhibit increased IL-1 and TNF-α mRNA and, therefore, contribute to ongoing neuroinflammation [75,76]. Dysfunctional astrocytes, in turn, may further contribute to the CSD burden via reversed glutamate transport from the ECF, wherein this neurotransmitter is released rather than absorbed [77].

### 6.3. Platelets

Platelets within the subarachnoid space are activated by numerous factors, including reactive microglia, dysfunctional endothelium, and other activated platelets [78,79] (See Figure 5). Activated platelets promote the endothelial release of von Willebrand factor and P-selectin expression, as well as the further release of inflammatory cytokines, creating a self-reinforcing inflammatory and procoagulant state [80]. Thus, microthrombi formation plays an increasingly recognized role in DCI via both vascular compromise and amplification of inflammatory response and glutamate release [2,9,81]. A procoagulant state has been noted to precede DCI, correlating with higher thromboxane levels [78]. In one small study including 12 aneurysmal SAH patients, a persistent increase in platelet reactivity was described [82]. Thus far, a meta-analysis has found no effective single antiplatelet agent to significantly prevent DCI [83]. This is likely due to the inflammation-promoting effects of platelets beyond aggregation.

### 6.4. Molecular Mediators of Neuroinflammation in SAH

#### 6.4.1. Interleukin 1

Interleukin-1 (IL-1) is a pleiotropic cytokine mainly produced by microglia and released by inflammasomes [84]. IL-1 receptors are expressed by neurons, astrocytes, microglia, and endothelial cells. It has a broad range of physiological and pathological effects on the brain [85]. At physiologically low levels, Interleukin 1 aids in synaptic pruning and plasticity, memory formation, and sleep [85,86]. In the setting of neurological insult, though, there is a pathological rise in IL-1 levels stimulated by neuroinflammatory cytokines. Elevated K+ concentrations act as a trigger for CSD and additionally activate inflammasomes such as NPLRP1 and NPLRP3 that release IL-1 [85,87]. Cortical spreading depolarizations have also been linked to triggering a proportional increase in IL-1 [88,89].

Elevated levels of IL-1 have been detected in the cerebral cortex and cerebrospinal fluids of humans in the setting of SAH [90,91]. These elevations of IL-1 subsequently promote the activation of matrix metalloproteinases (MMP) [92,93]. The consequential blood-brain barrier interruption paves the way for a neuroinflammatory response. IL-1 may play a role in microvascular dysfunction as it can promote sarcoplasmic Ca++ release, myosin light chain phosphorylation, and possibly stimulation of platelet-derived growth factor [94], thereby leading to vasoconstriction and contributing to NVU uncoupling.

IL-1 receptor antagonists are undergoing phase III trials [95].

#### 6.4.2. Interleukin 2

Interleukin-2 (IL-2) is a cytokine that is produced mainly by neurons [96]. Whereas many of the cytokines have proinflammatory properties, IL-2 indirectly decreases inflammation by driving T regulatory cell expansion [96]. These T-regulatory cells have been shown to have neuroprotective effects. Delivering low doses of IL-2 has been shown to reduce neuronal injury and decrease inflammation after TBI and SAH in murine models [96,97]. Conversely, elevated serum levels of IL-2 were shown to be positively correlated with poor prognosis following SAH [98]. More studies are needed to differentiate the role of IL-2 in the central nervous system and better define its therapeutic potential.

#### 6.4.3. Interleukin 6

Interleukin-6 (IL-6) is expressed predominantly by neuronal cells, microglial cells, and astrocytes, as well as endothelial cells [99] and is the most widely studied cytokine in SAH [100,101]. The active release of IL-6 occurs in the setting of complex mechanisms involving other cytokines. TNF-α, for example, acts via the NFκB pathway in astrocytes to promote IL-6 but fails to activate microglial cells in a similar fashion [102]. Conversely, GM-CSF stimulates microglial IL-6 but not astrocytes. Glutamate-mediated membrane depolarization is one of the main mechanisms for neuronal upregulation of IL-6 [103,104].

IL-6 may serve as an important biomarker for DCI [91,105,106]. IL-6 levels typically rise one week after the index hemorrhage and remain elevated for up to 14 days, thereby demonstrating a temporal correlation with the incidence of DCI. Individual CSF peak levels correlated significantly with DCI [106]. Additionally, the IL-6 levels during this time are disproportionately elevated in the CNS, indicating that its rise is not merely a systemic response to the hemorrhage [100]. Elevated IL-6 may further act as a predictor of poor outcomes or even death [106,107]. Currently, the implications of these studies are limited by a means to measure IL-6 directly, the absence of standardized measurement techniques, and the costs associated with the same.

#### 6.4.4. TNF-α

TNF-α is a pro-inflammatory cytokine produced by microglia and astrocytes. This cytokine has been associated with oxidative stress and neuronal apoptosis. It promotes neuroinflammation by increasing the expression of pro-inflammatory adhesion molecules [108,109]. As an added deleterious effect, TNF-α may promote neurovascular uncoupling via the downregulation of endothelial nitric oxide synthase [2,94]. It may also directly facilitate CSDs by promoting the expression of AMPA receptors in postsynaptic neurons [110,111].

Understanding of TNF-α and its effects is far from complete, as it was shown to decrease CSDs in rats, possibly via GABAergic properties [85]. Ultimately, higher TNF plasma and CSF levels have been associated with poor clinical outcomes and an increase in lesions found on MRI diffusion-weighted imaging, independent of radiologic vasospasm [108,112]. Based on current evidence, it can be concluded TNF-α possesses pleiotropic effects that are tipped toward causing harm in humans (Table 1).

### 6.5. Neuroinflammation and Vasoconstriction

Local inflammation promotes neurovascular uncoupling through numerous vasoactive substances that are released by astrocytes and endothelial cells [2]. 20-Hydroxyeicosatetraenoic acid is an arachidonic acid derivative that inhibits vascular K+ channels and thereby causes localized vasoconstriction [94,118,119,120]. Astrocytic-derived thromboxane prevents vascular dilation (Figure 6) [14]. Endothelin-1 (ET-1) promotes Ca++ entry into myocytes causing smooth muscle constriction [94]. ET-1 further stimulates cytokine and metalloproteinase production while inhibiting astrocyte glutamate transporters. In tandem, these measures reduce blood flow, impair CSF clearance, and disrupt recovery mechanisms. This is supported by murine models in which the ET-1 exposed cortex exhibited a higher CSD burden, with longer-lasting periods of subsequent electrical silence, suggesting delayed recovery [121].

## 7. Treatment of CSDs and Neuroinflammation in SAH

CSDs do not necessarily become injurious if downstream effects are diminished or flow is restored prior to the point of no return leading to cell death [122]. They represent a window of opportunity to prevent DCI. Potential treatments may target neuroinflammation, the inhibition of CSD genesis, or propagation. Although the initiation of CSDs are observed up to several hours and may persist for days after the initial insult [27,123], the window for effective neuroprotective strategies remains to be elucidated and may be narrower.

Management of DCI to date has focused on vasospasm as the main cause [8], with recommendations including a trial of hypertension (presuming the aneurysm is secured), maintenance of euvolemia, and intra-arterial vasodilators administered locally. However, guideline recommendations against prophylactic balloon angioplasty [124] again point towards more complex phenomena than vasospasm. Moreover, current treatments have shown incomplete efficacy. Herein, we focus on the interplay between inflammation and CSD in the context of existing and emerging therapeutic interventions (Table 2).

### 7.1. Nimodipine

Nimodipine is a dihydropyridine L-type Ca++ channel blocker and the only medication that has been proven to reduce the risk of DCI in SAH patients. It is the only Class 1, Level A medication in current SAH treatment guidelines [124]. It has been suggested nimodipine may cause a reversion to physiological NVU coupling response [125]. In murine models, this medication has been shown to reduce CSD burden [51,61] and delay cerebral ischemia [131]. Similar results have been shown in humans. In a phase-3 pilot trial enrolling five patients, of which three exhibited CSDs, treatment with local nimodipine led to lower CSD incidence and diminished electrographic depression time [132].

The exact pharmacotherapeutic mechanism differentiating nimodipine from the rest of the dihydropyridines remains to be fully elucidated. It has been posited that what differentiates nimodipine from other Ca++ channel blockers is the induction of endogenous fibrinolysis, which in turn may help reduce the burden of inflammation-promoted microthrombosis [51].

### 7.2. N-Methyl-D-aspartate (NMDA) Receptor Antagonists

Glutamate-mediated activation of NMDAr plays a key role in CSDs and neuroinflammation [33,35]. In theory, NMDA blockers should impede catastrophic rises in extracellular K+, ameliorating neurovascular uncoupling [133]. Thus, NMDA receptor antagonists have been extensively studied in preclinical settings.

#### Ketamine

Ketamine has been historically used as a sedative, but increasingly its effective neuroprotective potential has been recognized. The relative success of this medication, as compared to other NMDAr blockers, may be due to indirect anti-inflammatory properties that have been demonstrated experimentally in murine lipopolysaccharide-mediated microglia models [134]. In vitro studies demonstrated ketamine reduces circulating tumor necrosis factor-α, interleukin 6 (IL-6), and C-reactive protein levels [128,129].

Evidence for the use of ketamine in the management of CSDs in humans first emerged in 2009, wherein a 34-year-old man with SAH showed decreased CSD activity as measured by subdural electrodes [135]. One retrospective study included 43 patients with mixed etiologies (SAH, TBI, hemispheric stroke, intracerebral hemorrhage). In this cohort, suppression of beta frequencies diminished CSD incidence; the authors proposed this was partially ascribable to ketamine administration [136]. Another multicenter retrospective study included 31 patients with intracerebral hemorrhage (ICH) or SAH. The authors concluded that ketamine reduced both the incidence of CSDs and the number of CSD clustering events significantly, while midazolam exacerbated said phenomena. However, this study was limited by a lack of dosing standardization, with numerous dose adjustments even within the same patient [136].

Yet another retrospective cohort study included 66 SAH patients, 33 of whom received ketamine. The authors did find a statistically significant reduction in CSD burden in the ketamine arm after treatment was initiated. Furthermore, the authors noted a correlation between higher ketamine dosage and CSD amelioration, as well as no tolerance at 5 days [127]. The first prospective randomized trial included demonstrating 10 human patients suffering from severe Traumatic Brain Injury or SAH. Ketamine doses of 1.15 mg/kg/h or higher were associated with a lower likelihood of spreading depression when compared to lower dosages (OR = 13.838, 95% CI = 1.99–1000) [53,127,137].

Although ketamine is a promising agent, its current use in the prevention and treatment of CSDs is limited by the lack of a standard protocol.

### 7.3. Heparinoids

Heparin is a linear polysaccharide from the glycosaminoglycan family vastly employed for its anticoagulant properties. Beyond these well-recognized anticoagulant properties, heparinoids may bind inflammatory mediators and oxyhemoglobin, scavenge ROS, and decrease the vasoconstricting effects of endothelin (Figure 7). Thus, it is posited that heparin may decrease leukocyte entry into inflamed tissue [130].

During SAH, neuroinflammation is coupled with endothelial dysfunction and CSDs, ultimately leading to secondary microthrombi formation [15]. Due to its anti-inflammatory and anti-thrombotic properties, Heparin is emerging as a promising strategy [15,138].

One large study included 718 SAH patients, 197 of whom received heparin. The treatment arm exhibited reduced DCI. Patients receiving heparin also exhibited improved clinical outcomes 6 months post-ictus [15]. A smaller study included 117 SAH patients, with a treatment arm of 57 patients receiving enoxaparin (low-molecular-weight fractionated heparin). Again, the heparin arm exhibited a reduction in DCI and improved overall outcomes at 1 year. However, another randomized clinical trial contested these results [138].

### 7.4. Nitric Oxide Donors

The interaction between nitric oxide and CSDs is more complex than a neurovascular effect given nitric oxide upstream guanylate cyclase regulation of voltage-gated NMDArs [56].

Assays have attempted to address neurovascular uncoupling with agents such as nitric oxide and nimodipine [2]. Nitric oxide donors (L arginine) or nimodipine administration reverted spreading ischemia to spreading hyperemia, resulting in accelerated tissue recovery in rats [139]. Thus far, the administration of systemic nitric oxide donors has been limited by systemic hypotension; it has been suggested that inhaled nitric oxide donors may bypass this effect [11].

### 7.5. Antiepileptics

Lamotrigine stabilizes voltage-sensitive Na+ channels, preventing aspartate and glutamate release. It was shown to have neuroprotective effects in various animal studies [140,141,142,143,144] as well as in combination therapy in one clinical trial [145]. Topiramate has also been noted to decrease CSDs, likely secondary to Na+ channel modulation, while carbamazepine does not affect CSDs despite also being a Na+ channel blocker [146].

## 8. Future Research

CSDs play a key factor in DCI and the development of neuroinflammation. Increasing adoption of multimodality monitoring will permit real-time detection of this phenomenon and its consequences is likely to aid the discovery and development of more effective treatment strategies [147].

With this said, the invasive nature of measuring CSDs cannot be overlooked. Patients undergoing ECoG may represent a subgroup tending towards severely ill to the point of necessitating closer monitoring. There is room for future studies to explore the effectiveness of less invasive ways of measuring CSDs.

Future treatment avenues include decreasing glutamate efflux or increasing its clearance, selective NMDA blockade, or targeting downstream effectors, including apoptosis mediators. Selective NR2B subunit blockade is, therefore, a possible future treatment avenue [27,129]. Another approach is uncoupling NMDARs from their pro-apoptotic signaling effects or pathways [148,149,150,151]. Additionally, other possibilities include inflammatory modulation via interleukin antagonists or downstream regulators; notably, IL-1 antagonists have completed phase II trials.

## 9. Conclusions

CSDs may be predictable and measurable markers of DCI in SAH [25]. A summation of the mechanisms by which CSDs contribute to the pathophysiology of DCI can be found on Table 3.

Taking the above into consideration, we propose that there is ample experimental and clinical evidence to state DCI is a phenomenon more complex than mere vasospasm. DCI is the intersection of various mechanisms of secondary injury in large part mediated and sustained by CSDs and consequent derangements of tissular metabolism, vascular supply, and inflammation. Advances in monitoring technology have further shown CSDs are not a stereotyped phenomenon but rather a complex series of electro-pathological, neurovascular, and immunological events, which may offer various treatment targets. Early CSD detection in SAH may allow for better risk stratification and prediction of neurological outcomes in SAH patients. Prevention or timely treatment of CSDs in patients may improve outcomes and avert DCI.

An effective neuroprotective strategy would unlikely possess a single mechanism of action, given the complexity of the interplay between the various mechanisms which build up to the development of DCI.

## Figures and Tables

**Figure 1 ijms-24-09883-f001:**
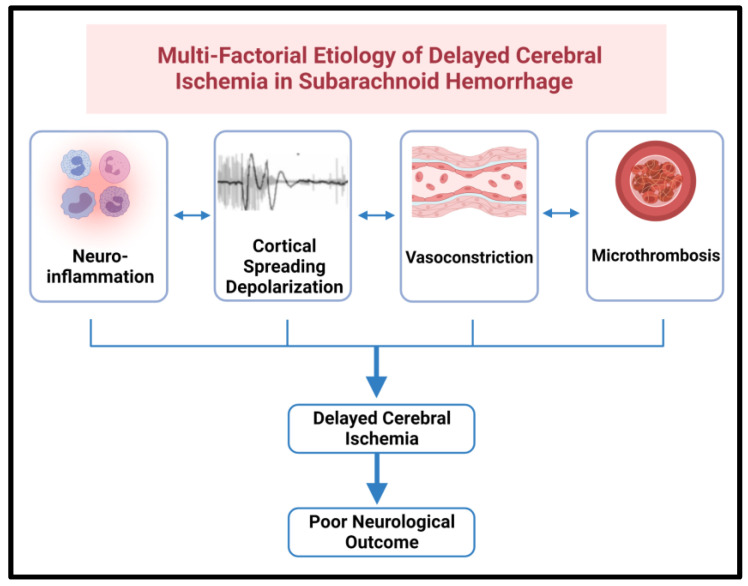
Delayed Cerebral Ischemia is a consequence of the interplay between neuroinflammation, cortical spreading depolarization, vasoconstriction, and micro thrombosis. Each of these phenomena can exacerbate the other and lead to DCI and poor neurological outcomes in patients with subarachnoid hemorrhage. CSDs lead to neuroinflammation, which may facilitate further CSDs. Further, CSDs and neuroinflammation contribute to supply-demand mismatch as well vasoconstriction and microthrombosis, respectively.

**Figure 2 ijms-24-09883-f002:**
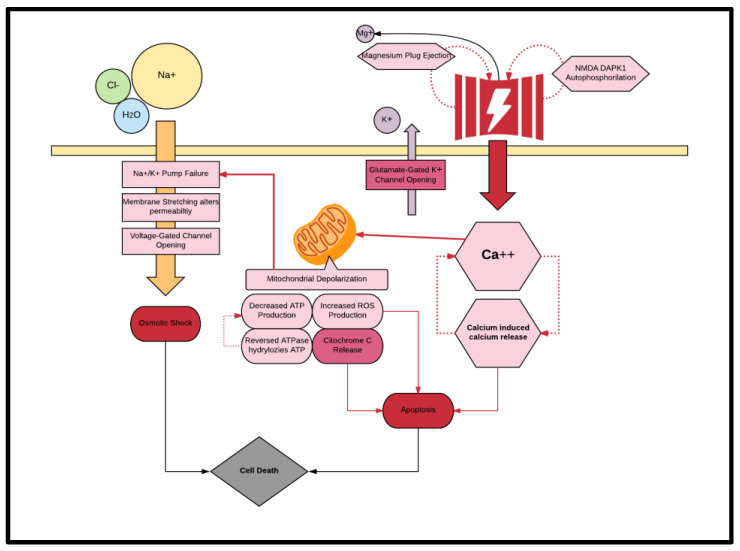
The above figure demonstrates the concepts of Cortical Spreading Depolarization and Excitotoxicity at a cellular level. Glutamate receptor activation at the neuronal membrane leads to mitochondrial depolarization and failure of the Na-K ATPase. The consequential disproportionate influx of sodium promotes a pathological increase in intracellular fluid and promotes cell death.

**Figure 3 ijms-24-09883-f003:**
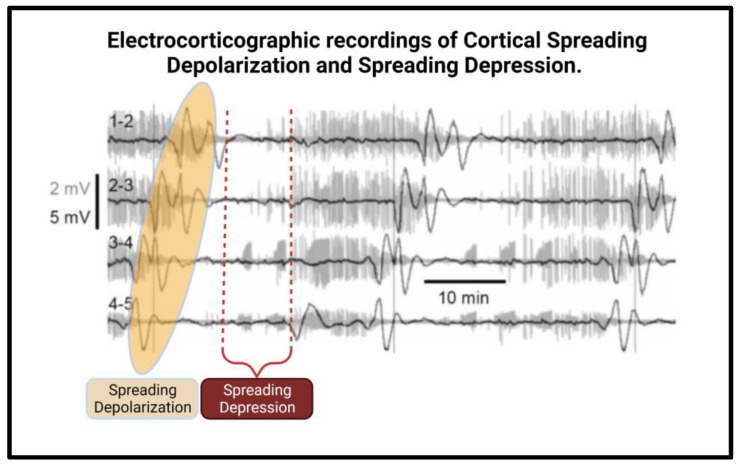
The above electrocorticographic records show slow (0.005–0.5 Hz, black) and Fast (0.5–30 Hz, grey) activity overlaid. Slow activity (highlighted in orange) shows spreading depolarization, which causes spreading depression (framed in red brackets) of fast activity.

**Figure 4 ijms-24-09883-f004:**
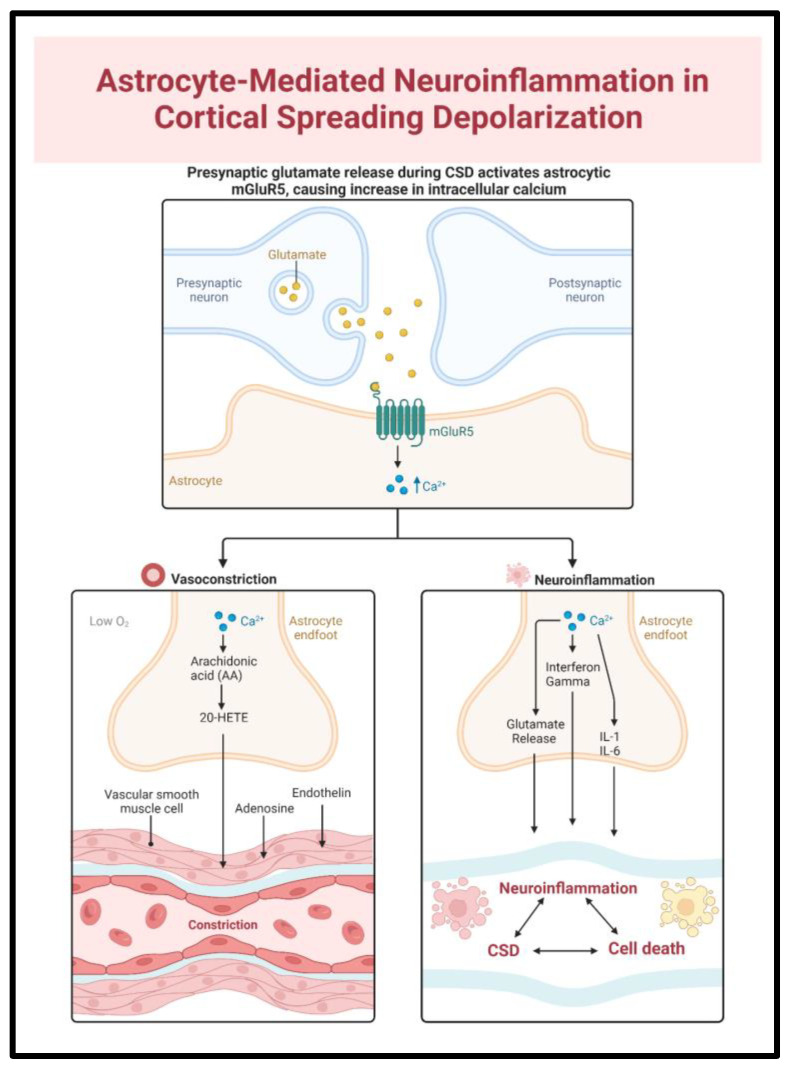
Subarachnoid hemorrhage-mediated astrocyte activation promotes both the release of vasoconstrictive and pro-inflammatory agents, playing a key role in the interplay of CSD, neuroinflammation, and vascular dysfunction. Activated astrocytes contribute to neuroinflammation via the release of IL-1, IL-6 and to further CSD via glutamate release.

**Figure 5 ijms-24-09883-f005:**
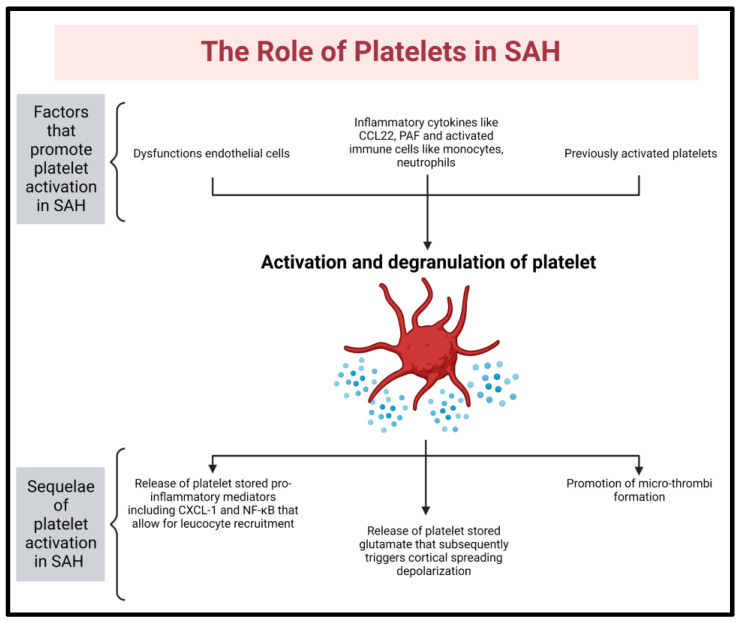
Platelets are activated by neuroinflammatory mediators. Upon activation, they release stores of glutamate and thereby further exacerbate CSDs. Additionally, they promote microthrombosis and neuroinflammation by contributing toward leucocyte recruitment [79].

**Figure 6 ijms-24-09883-f006:**
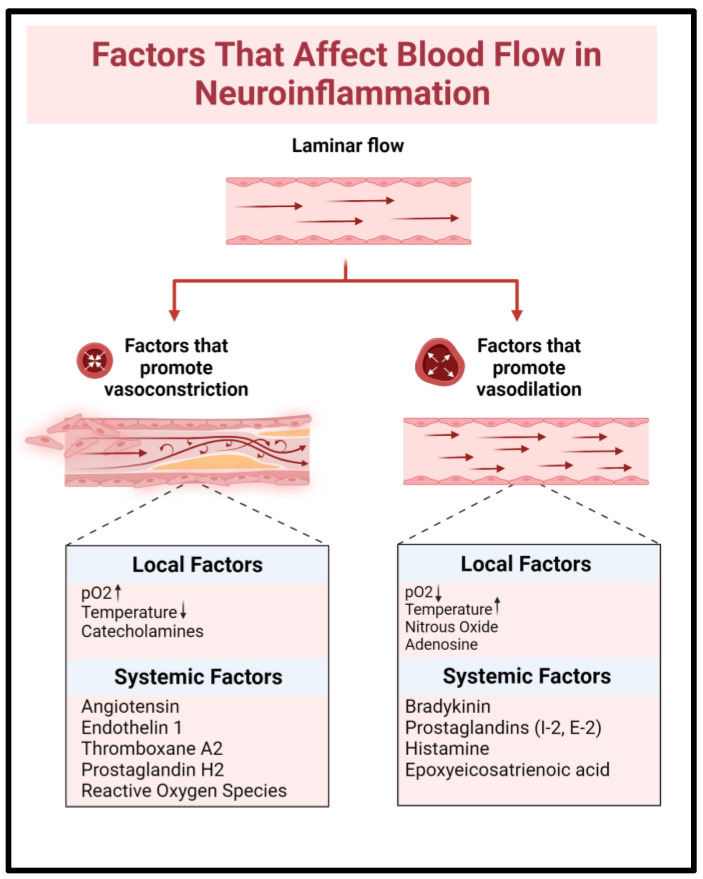
Proinflammatory mediators, including catecholamines, Angiotensin, and Endothelin 1, promote vasoconstriction in the setting of SAH-induced CSDs. ↑; increased, ↓; reduced.

**Figure 7 ijms-24-09883-f007:**
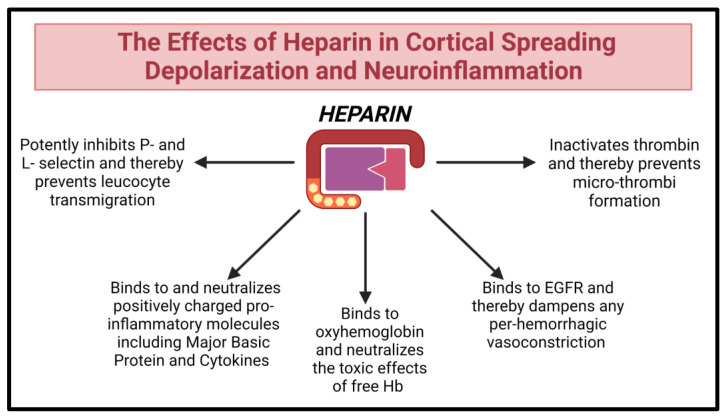
Although Heparin is well known for its anticoagulant properties, its anti-inflammatory effects make it a potential agent for amelioration of DCI in SAH.

**Table 1 ijms-24-09883-t001:** The Effects of Pro-Inflammatory Cytokines During CSDs and Neuroinflammation.

Interleukin	Source	Effects on CNS
IL-1	Produced by microglia [45]	Causes vascular inflammation, which leads to alterations in vascular tone, plaque formation, and reduced cerebral blood flow; exacerbates existing brain injury; leads to reductions in neurogenesis; pyrogenic [86]
IL-2	Produced by neurons [96]	Drives T regulatory cell expansion and protects against neuroinflammation [96]
IL-6	Produced by neurons, astrocytes, microglia, and endothelial cells [99]	Involved in neurogenesis; aids in the transition from innate to acquired immunity, recruiting monocytes and T cells; plays a role in the development of sensory neurons; regulates body temperature and energy expenditure [99]
TNF	Produced by microglia [113]	Causes tumor cell necrosis and apoptosis; mediates both acute and chronic neuroinflammation; regulates blood-brain barrier permeability, fevers, glutamate transmission, and synaptic plasticity [113]
ICAM-1	Expression in microglia, astrocytes, and endothelial cell [114]	Facilitates the adhesion and migration of neutrophils and monocytes [115]
IFN	Produced by microglia, astrocytes, and neurons [116]	Regulates homeostasis by removing myelin debris and limiting the permeability of the blood-brain barrier. [117]

**Table 2 ijms-24-09883-t002:** Prospective Therapies against SAH-induced CSDs and Neuroinflammation.

Treatment	Effect on CSDs	Effect on Inflammation	Vascular Effects (Vasoconstriction and Microthrombosis)
Nimodipine	Lowers CSD incidence and reduces the duration of electrographic depression through an unclear mechanism [51,61]	Yet to be determined	May decrease microthrombosis through its fibrinolytic capabilities [51] and possibly reverses neurovascular coupling [125]
Ketamine	Reduces the frequency, amplitude, and propagation of CSDs [53,126,127]	Reduces circulating tumor necrosis factor-α, interleukin 6 (IL-6), and C-reactive protein levels [128,129]	Reduction of circulating TNF-α, interleukin 6 (IL-6), and C-reactive protein levels [128,129]
Heparin	Yet to be determined	May bind pro-inflammatory mediators, including cytokines, DAMPs, P and L-selectins, and thereby reduce leukocyte burden [130]	Ameliorate micro thrombosis burden through its anticoagulant properties [15]

**Table 3 ijms-24-09883-t003:** Various Mechanisms by which CSD can induce DCI have been discovered [7,152].

CSDs drastically increase the energy expenditure of involved tissue. Recovery from CSD also incurs an energy deficit due to dependence on Na-K ATPase.
CSDs may be associated with reductions in perfusion in the setting of increased metabolic demand (neurovascular uncoupling). The resultant oligemia, in turn, increases extracellular K and thereby further promotes vasoconstriction and ischemia.
CSD independently promotes an intense inflammatory response and breakdown of the blood-brain barrier.

## Data Availability

No new data was created or analyzed in this study. Data sharing is not applicable to this article.

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
