# Peer review of "Cortical Spreading Depolarization and Delayed Cerebral Ischemia; Rethinking Secondary Neurological Injury in Subarachnoid Hemorrhage"

_ijms, 2023, doi:10.3390/ijms24129883_

Round 1
Reviewer 1 Report
In this review manuscript the authors discussed the role of cortical spreading depolarization in the condition of DCI/SAH. Potential therapeutic agents were also mentioned. Some concerns and suggestions are listed as below:
In line 11, please provide the full name of SAH.
In Figure 1, potential relationship between neuroinflammtion and CSD was not clear. How about cellular and molecular mechanisms?
In line 66, why Spanish articles were included? How many?
In line 70, 'Cortical spreading depolarizations (CSDs)' should be 'CSDs' since it had been mentioned in line 51.
In line 152, 'Sah' should be 'SAH'.
In the section of neuroinflammation (lines 200-219), it is not clear for readers if these cytokines were produced by microglia. Microglia may play different roles during different disease stages. Again, the role of glial cells in CSD was not discussed in details.
In line 214, how did you define 'microglial activation'?
In line 218, the authors said 'microglia suppression via PLX3397 decreased microglia cell accumulation. In fact, experimental microglial depletion using CSF1R inhibitor PLX3397 exert crucial influences on circulating monocytes and peripheral tissue macrophages, suggesting that effects on peripheral immunity should be considered both in interpretation of microglial depletion studies.
How astrocytes contribute to CSD burden? This was not discussed in details in the manuscript.
Overall, some statements are not accurate in this review paper. This is a major concern.
general
Author Response
In line 11, please provide the full name of SAH.
We have provided the full name for Subarachnoid Hemorrhage
In Figure 1, the potential relationship between neuroinflammation and CSD was not clear. How about cellular and molecular mechanisms?
Expounded on said mechanisms
In line 66, why were Spanish articles included? How many?
FG’s native language is Spanish, no Spanish article was included in the final paper.
In line 70, 'Cortical spreading depolarizations (CSDs)' should be 'CSDs' since it had been mentioned in line 51.
Corrected
In line 152, 'Sah' should be 'SAH'.
Corrected
In the section of neuroinflammation (lines 200-219), it is not clear for readers if these cytokines were produced by microglia. Microglia may play different roles during different disease stages. Again, the role of glial cells in CSD was not discussed in details.
Cytokine table explains the origin of each cytokine in detail
In line 214, how did you define 'microglial activation'?
The role of glial cell activation and definition was added, one new citation was included.
In line 218, the authors said 'microglia suppression via PLX3397 decreased microglia cell accumulation. In fact, experimental microglial depletion using CSF1R inhibitor PLX3397 exert crucial influences on circulating monocytes and peripheral tissue macrophages, suggesting that effects on peripheral immunity should be considered both in interpretation of microglial depletion studies.
The authors have included a line regarding the effects of PLX3397 on peripheral immunity
How astrocytes contribute to CSD burden? This was not discussed in details in the manuscript.
We have further expounded on astrocyte contribution to CSD burden.
Reviewer 2 Report
The present review is an interesting study linking cortical spreading depolarization (CSD), emerged after Subarachnoid Hemorrage, with the apparition of delayed cerebral ischemia (DCI). This review brings a good summary of the multifactorial events leading to CSD, its consequences, and the cellular and molecular pathways that link it to DCI. There are only minor concerns that should be addressed before publication:
- There are several acronyms that should be described somewhere, such as TCD, NOP, NMDA, MRI, EEG, etc.
- Pg 2. Line 64. Writting error: “neuroin“spreading depolarization”
- Pg 5. Line 173. Cubic mm should be more understandable as mm3
- Pg 6. First paragraph. These sentences: The authors measured cerebral circulation time with angiography and divided it into proximal and peripheral groups at 7-11 days from onset. Multivariate analysis showed the number of CSDs on the day of angiography was the only significant factor.¨ are not understandable. Please, describe it better.
- Figure 4. This is a nice figure showing the effect of astrocyte end-feet on the vessels. It would be good that the molecular pathways that are shown in the figure were explained somewhere. A minimum description would be enough, even in the figure legend.
- Table 1. If IL1, Il2, IL6 and TNFa are explained in the main text, a few lines regarding ICAM and IFN would make the table 1 more understandable also.
- In section 6, 20-Hydroxyeico-satetraenoic acid is mentioned, whereas in figure 6 is mentioned epoxyeicosatetraenoic acid. Would be good to mention the same acid.
- In section 6.3, it is mentioned that there is no effective single antiplatelet agent to significantly prevent DCI, whereas in section 7.5 it is mentioned that heparin is effective at reducing DCI. Heparin is an anticoagulant and antiplatelet agent. Please, modify it accordingly.
Author Response
- There are several acronyms that should be described somewhere, such as TCD, NOP, NMDA, MRI, EEG, etc.
Spelled out as requested
- Pg 2. Line 64. Writing error: “neuroin“spreading depolarization”
Corrected
- Pg 5. Line 173. Cubic mm should be more understandable as mm3
Corrected
- Pg 6. First paragraph. These sentences: The authors measured cerebral circulation time with angiography and divided it into proximal and peripheral groups at 7-11 days from onset. Multivariate analysis showed the number of CSDs on the day of angiography was the only significant factor.¨ are not understandable. Please, describe it better.
Rewritten as suggested
- Figure 4. This is a nice figure showing the effect of astrocyte end-feet on the vessels. It would be good that the molecular pathways that are shown in the figure were explained somewhere. A minimum description would be enough, even in the figure legend.
Included a summary description in the figure legend
Table 1. If IL1, Il2, IL6 and TNFa are explained in the main text, a few lines regarding ICAM and IFN would make the table 1 more understandable also.
Included a few lines in the main text regarding ICAM and IFN before said table
- In section 6, 20-Hydroxyeico-satetraenoic acid is mentioned, whereas in figure 6 is mentioned epoxyeicosatetraenoic acid. Would be good to mention the same acid. The mentioned acids are different molecules forming part of differing effector arms.
- In section 6.3, it is mentioned that there is no effective single antiplatelet agent to significantly prevent DCI, whereas in section 7.5 it is mentioned that heparin is effective at reducing DCI. Heparin is an anticoagulant and antiplatelet agent. Please, modify it accordingly. Heparin is considered an anticoagulant agent, furthermore the effects proposed are due to anti-inflammatory effects of said drug.
Reviewer 3 Report
This comprehensive article offers an in-depth analysis of the current knowledge surrounding cortical spreading depolarization (CSD) in subarachnoid hemorrhage, including potential therapeutic interventions. The authors present a thorough review of existing literature and provide insightful analysis of the pathophysiology of CSD and its impact on delayed cerebral ischemia (DCI), as well as the pharmacotherapeutic mechanisms of various treatments. It is a valuable resource for clinicians and researchers interested in the field. However, to provide a more complete picture, the authors could address the limitations associated with the invasiveness of detecting CSDs. Specifically, it could be noted that the patients who underwent ECoG measurements in the studies discussed may represent a subgroup with a higher likelihood of developing DCI and poorer outcomes.
Author Response
The reviewer makes an EXCELLENT point, we have added this line of thought as limitations under the future research section.
Round 2
Reviewer 1 Report
The authors have addressed my concerns.
fine